# Cancer of Unknown Primary: Challenges and Progress in Clinical Management

**DOI:** 10.3390/cancers13030451

**Published:** 2021-01-25

**Authors:** Noemi Laprovitera, Mattia Riefolo, Elisa Ambrosini, Christiane Klec, Martin Pichler, Manuela Ferracin

**Affiliations:** 1Department of Experimental, Diagnostic and Specialty Medicine (DIMES), University of Bologna, 40126 Bologna, Italy; noemi.laprovitera@unibo.it (N.L.); mattia.riefolo@unibo.it (M.R.); elisa.ambrosini3@studio.unibo.it (E.A.); 2Department of Life Sciences and Biotechnologies, University of Ferrara, 44121 Ferrara, Italy; 3Division of Oncology, Medical University of Graz, 8036 Graz, Austria; christiane.klec@medunigraz.at (C.K.); martin.pichler@medunigraz.at (M.P.)

**Keywords:** cancers of unknown primary site, primary site identification, clinical management, molecular profiling, liquid biopsy

## Abstract

**Simple Summary:**

Patients with cancer of unknown primary site suffer the burden of an uncertain disease, which is characterized by the impossibility to identify the tissue where the tumor has originated. The identification of the primary site of a tumor is of great importance for the patient to have access to site-specific treatments and be enrolled in clinical trials. Therefore, patients with cancer of unknown primary have reduced therapeutic opportunities and poor prognosis. Advancements have been made in the molecular characterization of this tumor, which could be used to infer the tumor site-of-origin and thus broaden the diagnostic outcome. Moreover, we describe here the novel therapeutic opportunities that are based on the genetic and immunophenotypic characterization of the tumor, and thus independent from the tumor type, which could provide most benefit to patients with cancer of unknown primary.

**Abstract:**

Distant metastases are the main cause of cancer-related deaths in patients with advanced tumors. A standard diagnostic workup usually contains the identification of the tissue-of-origin of metastatic tumors, although under certain circumstances, it remains elusive. This disease setting is defined as cancer of unknown primary (CUP). Accounting for approximately 3–5% of all cancer diagnoses, CUPs are characterized by an aggressive clinical behavior and represent a real therapeutic challenge. The lack of determination of a tissue of origin precludes CUP patients from specific evidence-based therapeutic options or access to clinical trial, which significantly impacts their life expectancy. In the era of precision medicine, it is essential to characterize CUP molecular features, including the expression profile of non-coding RNAs, to improve our understanding of CUP biology and identify novel therapeutic strategies. This review article sheds light on this enigmatic disease by summarizing the current knowledge on CUPs focusing on recent discoveries and emerging diagnostic strategies.

## 1. Introduction

The development of metastases represents the main cause of cancer-related deaths and affects patients’ response to therapy and life expectancy [1]. A difficult to manage scenario of a metastatic disease is represented by cancers of unknown primary sites (CUPs), a heterogeneous group of metastatic tumors whose primary site of origin cannot be determined at the time of diagnosis, even after comprehensive clinical and pathological investigations [2].

CUP cases share some common features, such as early dissemination, aggressiveness, poor prognosis, and an unpredictable metastatic pattern [3]. Most CUP patients (~80%) do not respond to chemotherapy and have a median overall survival of 6 to 10 months, but a fraction of all cases, about 20%, are characterized by a favorable prognosis due to their potential chemosensitivity and long-term disease control [2,4,5].

Even today, it is difficult to provide incidence data about CUPs. Historically, CUPs were reported as 3–5% of novel worldwide cancer diagnoses [2]. However, in the last few years CUPs, have decreased to 1–2% of all cancer cases [6], but still rank among the ten most common causes of cancer-related deaths [7,8,9]. The decreasing trend of CUP incidence over the last few decades (reviewed in ref 6) can be attributed to the gradual improvement of diagnostic techniques, and it is therefore challenging to compare incidence and outcome studies from different decades. Moreover, socioeconomic disparities between countries and the unequal access to diagnostic workup and treatment seem to impact considerably on CUP incidence [10].

The hypothesis of a “familial CUP” syndrome is also a matter of debate. Familial clustering studies reported an increased risk of CUP in first-degree relatives, which is also associated with the occurrence of other types of tumors (lung, kidney, liver, ovarian, colorectal, breast, and melanoma) [11,12]. Other risk factors identified so far include smoking habits [8] and human papillomavirus (HPV) infection for squamous cell CUPs [13,14].

CUP origin and underlying biology remains an enigma, and different theories were developed by the scientific community about the possible origin of CUPs: some researchers suggested that CUPs originate from small undetectable, dormant, or later regressed primary lesions; others asserted that it is necessary to abandon the traditional tissue-gnostic approach and consider CUPs as early disseminating, aggressive, metastatic tumors with no existent primary site [15]. However, when post-mortem autopsies were investigated, a primary site was discovered in 70% of CUP cases, with the lung and pancreas as the most frequently detected organs of origin [16].

Riding the wave of tissue-gnostic approach, several tissue-of-origin classifiers have been developed, collecting evidence that supported their translational potential in the clinical management of CUP patients. However, the advent of precision medicine and the availability of multi-omics databases is rapidly transforming the clinical management of different cancer types, shifting to a more personalized treatment approach, regardless of the primary site.

Here, we review the current knowledge on CUP research, focusing on patients’ current management, treatment options, and the latest approaches on CUP diagnostics based on molecular profiling and epigenetic characterization of the tumor.

## 2. CUP Diagnosis

When dealing with a potential CUP diagnosis, clinical practice guidelines suggest a thorough diagnostic workup [2,4], which includes routine clinical evaluation, extensive physical examination, blood/biochemical analyses, and radiological tests (chest radiography and computed tomography (CT) for the thorax, abdomen, and pelvis). If this initial series of diagnostic tests is still insufficient to identify the site of the primary tumor, additional specific tests can be considered. Depending on metastases localization, patient’s clinical signs, symptoms, and gender, these tests include magnetic resonance imaging (MRI), endoscopy, positron emission tomography (PET), or the assessment of specific serum tumor markers. Finally, the last and most critical step is represented by immunohistochemical testing, which remains the most important diagnostic tool in establishing the tissue of origin. If, after these attempts, the primary tumor remains elusive, a diagnosis of CUP is confirmed. This diagnosis is reversible; in fact, in some cases a primary tumor is revealed later during the course of the disease.

Post-mortem autopsy is performed only in a small percentage of patients. According to twelve cohort autopsy studies—five published between 1977–1980 [17,18,19,20,21] and seven between 1981–2005 [22,23,24,25,26,27,28]—the most common primary sites identified post-mortem in CUP patients were the lung (27%) and pancreas (24%), followed by the hepatobiliary tree (liver and bile duct) (8%), kidneys or adrenals (8%), colon (7%), and genital organs (prostate 2%, testis 0.3%, ovary/uterus 4%, and uterine cervix 0.7%) [16].

Moreover, many factors could lead to a misdiagnosis of CUP, such as a rushed premature diagnosis, the inexperience of the pathologist, or the scarce availability of biological material that limits the number of performable diagnostic tests.

### 2.1. Histological and Immunohistochemical Evaluation

Once the biopsy is obtained and the malignancy is clinically assumed, the pathologist uses a stepwise approach to identify the broad tumor type, the tumor subtype, and, whenever possible, the site of origin. CUPs are prevalently carcinomas; therefore, the first task of the pathologist is to evaluate, using hematoxylin and eosin-stained slides and routine light microscopy, the basic form of differentiation. Most CUP histologies are adenocarcinoma (60%) or poorly differentiated carcinoma (30%), but they can also be squamous (5%) or neuroendocrine carcinoma (5%) [29].

The underlying differences between distinct tissues depend on their gene expression profile: some genes are ubiquitously expressed and involved in basic cellular functions, others are expressed by one or few tissue types, determining their function and differentiation [30]. It has been demonstrated that the expression of a subset of genes is retained in metastatic cells and can be used to trace back the tissue of origin. However, this condition is better preserved in well-differentiated rather than poorly or undifferentiated cancers [31]. Even though the expression of a particularly informative gene might be lower in metastases compared to primary tumors, it could provide a decisive clue to identify the tissue of origin; for this reason, immunohistochemical (IHC) testing covers an essential role in pathological evaluation of metastases.

The first IHC testing is made to exclude tumor types with specific treatment requirements, such as lymphoma, melanoma, germ-cell tumors, or sarcoma. When dealing with a carcinoma, a first panel of 19 antibodies is assessed, followed by other additional tests if the previous were inconclusive. Despite the existence of specific practice guidelines for CUP IHC testing and the definition of diagnostic algorithms to identify the tissue of origin [4], small pieces of biopsy tissue limit this diagnostic workup. In fact, when the available biological material is scarce, pathologist’s intuition and experience play a pivotal role in markers selection. For this reason, one of the major limits of this technique is the lack of objective evaluation parameters. Moreover, this process can be time-consuming, complex, and can lead to ambiguous classifications, especially when dealing with poorly differentiated tumors. Different cancer types can also share the expression of the same markers, making it challenging to identify a univocal primary site.

Once the epithelial origin is established, the expression of two keratins (previously referred as cytokeratins), K7 and K20, which broadly define the subsets of carcinoma, is mostly used for CUP primary site predictions [32,33]. Belonging to the intermediate filament family, keratins are expressed in epithelial cells and derive from 54 functional genes in the human genome [34]. While K7 is found in some simple epithelia, K20 is expressed in a restricted set of epithelia (Table 1). However, beyond this general rule, some carcinomas are known for their lack of immunophenotypes, i.e., pancreatic carcinoma generally show K7^+^/K20^+^, even though it can lose K7; gastric adenocarcinoma can show all K7/K20 phenotypes; and cholangiocarcinoma can overlap with pancreatic carcinoma [35,36,37].

Once the four K7/K20 expression patterns (K7+/K20+; K7+/K20−; K7−/K20+; K7−/K20) are established, IHC testing can proceed with site-specific markers. These can be cytoplasmic and/or membranous markers of differentiation or nuclear transcription factors. The expression level and positive stained fraction of tumor cells for these cytoplasmic markers generally inform on the state of tumor differentiation: poorly differentiated tumors will be characterized by fewer positive cells compared with well-differentiated tumors. In contrast, nuclear transcription factors, when positive, are expressed in the entire tumor cell population, and expression is generally not associated with tumor differentiation.

Here, we describe the most informative site-specific markers.

#### 2.1.1. CDX2

The most well-known marker for metastatic carcinomas of gastrointestinal (GI) origin is CDX2 (Caudal Type Homeobox 2), a nuclear transcription factor that plays a role in intestinal epithelial cells proliferation and differentiation [38,39]. Virtually all colorectal adenocarcinomas are positive for CDX2, with relative loss of expression in microsatellite unstable subtype [40]. CDX2 is also expressed in neuroendocrine tumors of the GI tract [41,42]. The expression pattern is important for diagnosis, in fact while colorectal cancers (CRC) shows uniform and characteristic positive stain, other site carcinomas have variegated or focal patterns of expression [43]. In addition, in neuroendocrine tumors of the GI tract the intensity is generally much weaker and focal [44,45,46].

CDX2 is also expressed in gastric, pancreato-biliary tract carcinomas and in all carcinomas with histological colorectal appearances, such as ovarian mucinous [47,48], bladder [49], sinonasal [50], enteric subtype of mucinous, and non-mucinous pulmonary adenocarcinomas [46]. CDX2 can be expressed in neuroendocrine carcinomas of pancreas and non-GI tract high-grade neuroendocrine carcinomas, such as those of the bladder and lung [44,45,51].

Another marker for gastrointestinal origin is SATB2 (special AT-rich sequence-binding protein 2) that seems to be able to distinguish ovarian mucinous primaries from lower gastrointestinal metastases [52].

#### 2.1.2. GATA3

GATA binding protein 3 (GATA3), member of a zinc finger transcription factor family, is crucial for the differentiation of many tissues and a very sensitive marker for breast and urothelial carcinomas [53,54,55,56]. It could be expressed in endometrial, pancreatic, salivary gland, trophoblastic germ cell neoplasms, paragangliomas and mesotheliomas [57,58].

Other markers for breast origin are gross cystic disease fluid protein 15 (GCDFP-15) [59,60] and mammaglobin A (SCGB2A2) [61,62].

#### 2.1.3. TTF1

Thyroid transcription factor 1 (TTF-1), a 38 kDa member of the NKX2 family of DNA-binding transcription factors, is selectively expressed during the embryonic development of the thyroid, the diencephalon, and in the respiratory epithelium [63]. TTF-1 is expressed by neuroendocrine and non-neuroendocrine carcinomas of the lung [64]. Sensitivity is higher among adenocarcinomas and non-mucinous carcinoma, great in small cell carcinomas, while lower in large cell neuroendocrine carcinoma. TTF-1 expression, in fact, is not considered as specific for high-grade neuroendocrine carcinomas of lung origin [65]. TTF-1 expression was also found in small subsets of ovarian [66,67,68], endometrial [69], colorectal [70,71], and breast [72] carcinomas. Moreover, its expression was assessed in variable subsets of small cell carcinomas of genitourinary, gynecological [73,74,75,76] and dermatologic (Merkel cell) tumors [77,78].

#### 2.1.4. Napsin A

Another marker referable to lung origin is Naspin A, an aspartic protease for the maturation of surfactant B, that can be found type 2 pneumocytes and alveolar macrophages cytoplasm. The specificity of TTF-1 and Napsin A co-expression is extremely high for pulmonary adenocarcinomas [79,80]. However, Napsin A can also be identified in renal cell carcinomas, endometrial adenocarcinomas, papillary thyroid carcinomas, and clear cell carcinoma of ovary [81,82].

#### 2.1.5. PAX8

PAX8 (Paired Box 8) is a transcription factor that is critical to the embryonic development of the thyroid gland, kidney, and müllerian system. PAX8 is expressed in non-ciliated, mucosal cells of the fallopian tubes, endocervix, endometrium, and simple ovarian inclusion cysts, but not on the surface of the epithelial cells of the ovary [83,84]. PAX8 shows a high level of expression in non-mucinous ovarian carcinomas; in contrast, mucinous carcinomas of the ovary show lower levels of expression and, when positive, are typically focal.

PAX8 is highly expressed in endometrioid adenocarcinomas, uterine serous carcinomas, and endometrial clear cell carcinomas. Expression of PAX8 in the setting of invasive cervical adenocarcinomas is less well studied, with only a few reported as positive [85]. Studies have also shown that PAX8 is not expressed in mammary carcinomas.

## 3. CUP Classification and Treatment

International guidelines for tumor treatment are essentially based on primary site definition. With no evidence of a primary site, CUP patients cannot be treated with site-specific therapy; thus, they are managed based on their clinicopathological characteristics.

According to these criteria, they are classified into two prognostic subgroups, defined as favorable (15–20%) and unfavorable CUPs (80–85%). Favorable CUPs present common features with a specific known tumor type, sharing its metastatic pattern, response to treatment, and prognosis. These patients are usually treated in line with similar tumor types, show chemosensitivity and present a potentially curable disease. In fact, their life expectancy is significantly longer (15–20 months) and long-term disease control is achieved in 30–60% of cases [86]. CUP patients with cervical lymph node metastases from squamous cell carcinoma or peritoneal carcinomatosis from papillary serous carcinoma belong to this subgroup [86]. On the other hand, most CUPs belong to the unfavorable group and have a median survival of 6–10 months [87]. The prognostic model developed by the French CUP Group (GEFCAPI) further classify unfavorable CUPs on the basis of two parameters: the performance status and the pre-treatment serum lactate dehydrogenase levels (LDH) [88].

The subgroup characterized by a good performance status (0–1) and normal LDH levels when treated with a platinum-based doublet chemotherapy regimen (in combination with gemcitabine or taxane) experiences a median survival of 12 months [2,89]. The subgroup presenting a performance status ≥2 and elevated LDH levels is characterized by a dismal median survival of four months; for these patients, the situation is critical and because the primary goal for them is to improve their quality of life, treatment mainly consists of palliative care, symptom control, or low-toxicity chemotherapy regimens [4,5].

A meta-analysis evaluated the impact on CUP survival of platinum, taxane, or new-generation compounds (gemcitabine, vinca alkaloids or irinotecan) regimens, in monotherapy or in combination; of note, no chemotherapy agent was found to effectively prolong CUP patient survival [90]. In the most recent NCCN guidelines, 11 chemotherapy regimens are listed as indicated for adenocarcinoma and nine for squamous histology [4]. However, these guidelines remain empirical, because the evidence is mostly based on single-arm phase II clinical trials [91,92,93] or smaller trials [94,95,96].

An indication of a site of origin in these patients, or an approach based on personalized medicine, may assist the physicians in the selection of the best treatment options, potentially improving CUPs prognosis and survival.

## 4. Molecular Prediction of the Primary Site

It has been recently hypothesized that primary-oriented specific treatments would significantly improve CUP patients’ prognoses.

Guided from this assumption, different approaches based on mRNA, microRNA, or DNA methylation analysis were developed. The underlying premise for these molecular profiling assays is that, independently of the technology used, metastatic tumors have molecular signatures known to be tumor-specific and retained during metastatic processes. Although based on different multi-feature algorithms, these molecular predictors are developed using similar approaches: first the analysis of a cohort of tumors with a known primary site (reference or training set) to create a cancer-specific signature database that is employed as a tissue-of-origin classifier; then, based on the similarities with the tumor types included in the reference set, a tissue-of-origin is assigned to each metastatic tumor with uncertain/unknown origin.

The accuracy of each molecular assay can be directly estimated by analyzing the same primary tumors used to create the classifier—independent datasets of primaries or metastatic tumors of known origin. In general, molecular profiling tools declared an overall prediction accuracy that ranges between 80–95%, reflecting the great potential of these tools to assist and improve the diagnostic workup of CUP patients [97].

Many of these studies carry limitations: a low number of cancer-types or samples for each class in the reference set; the excessive cost for large-scale analyses (microarray-based studies); too few analyzed targets; technical failure analysis for low quality mRNA (GEP-based studies); or possible contamination of the tumor biopsy with surrounding tissues that could confound results.

However, it is challenging to verify the prediction consistency and overall performance in CUP site-of-origin predictions, due to the rarity of post-mortem autopsy or its failure in detecting the primary. Sometimes, the origin site is identified during the course of the disease, but in most cases, the prediction is evaluated using available surrogate measures (comparison with IHC findings, clinical presentation, and response to therapy). To date, randomized, prospective clinical trials have failed in demonstrating a clinical benefit for CUP patients from the use of molecularly directed therapies; therefore, current international and national guidelines do not recommend a routine use of tissue-of-origin predictors in CUP management, only a case-by-case evaluation [4,97].

### 4.1. Predictive Assays Based on Coding RNAs

Guided by the goal to overcome the limits of current cancer classification and establish a more precise molecular diagnosis, several researchers developed gene expression profile (GEP)-based molecular tools to classify human cancers [98,99,100]. Later, several molecular tests were developed and commercialized, such as the Pathwork Tissue of Origin Test (Pathwork Diagnostics, Redwood City, CA, USA), with a potential application in tissue-of-origin prediction of metastases of known origin [101,102,103]. In these microarray-based studies, the prediction accuracy was about 85–95%, both using frozen and formalin-fixed paraffin-embedded tissue (FFPE) samples, with poorly differentiated and undifferentiated primary tumors responsible for lower accuracy rates.

In a blinded, multicentric study, GEP prediction accuracy was compared with standard IHC for metastases of known origin diagnosis. When the diagnosis was achieved in a single round of IHC stains, IHC and GEP performance were similar, showing an accuracy higher than 90%; on the other hand, in those cases that required a second round of stains, GEP proved to be more accurate than IHC (91% versus 71%; *p* = 0.023) in determining the appropriate primary site [104].

When applied to CUPs in retrospective and prospective settings, GEP studies reported a consistency with clinical evaluation and pathological assessment that ranged from 62.5 to 85% [105,106,107,108], as shown in Table 2.

However, because microarray is a complex, time-consuming, and expensive technology, RT-qPCR was evaluated as an easier-to-use, rapid, and less expensive option. For this reason, several research groups used the available microarray data to identify focused signatures of small number of genes to discriminate between different cancer types, thus allowing a more focused analysis.

An example is represented by the 92-gene signature identified by Ma et al., able to distinguish and classify 26 cancer types [109] with an overall sensitivity of 87% for tumor types and 82% for subtypes [110]. This assay, commercially available since 2013 as CancerTypeID (bioTheranostics, San Diego, CA, USA), is a diagnostic tool used in the U.S.A. to solve uncertain cases, and proved to have a higher prediction accuracy than standard IHC (79% versus 69%) [111,112]. Moreover, another signature of 10 genes was used to design a qPCR assay by Veridex, able to efficiently discriminate six tumor types (lung, breast, colon, ovary, pancreas, and prostate) [113]. Notably, when these assays were applied to predict the CUP site of origin, predictions showed an agreement with the primary site later identified and/or the clinicopathological features of the tumor in 54–86% of cases (Table 2) [113,114,115,116].

In one study, a machine learning method called SCOPE (supervised cancer origin prediction using expression) was proposed as a diagnostic tool able to incorporate information from whole-transcriptome RNA sequencing data and solve complex diagnoses. The training step was performed on public datasets, including The Cancer Genome Atlas (TCGA) (*N* = 10,688 samples of 40 untreated primary tumor types and 26 adjacent normal tissues), while the testing was performed retrospectively on untreated primary mesothelioma (*N* = 211) and treatment-resistant metastatic cancers (*N* = 201). SCOPE was able to achieve 99% accuracy on mesotheliomas, and 86% for the sarcomatoid subgroup. This approach was also tested on 15 CUPs, with results in accordance with conventional pathology in 80% of cases [117].

Few clinical trials and case reports have evaluated the impact of molecular profiling on CUP patients’ survival when treated with site-specific regimens according to molecular predictions [112,115,118,119,120]. In a retrospective setting, among 42 CUP patients predicted as colorectal cancer with high probability, 76% were treated with colorectal cancer regimens and experienced a median survival of 27 months, significantly longer than historical reports (8–11 months) [116]. In another similar study, 7 of 25 GEP-profiled CUP patients were treated with site-specific therapy according to molecular prediction; of note, these patients showed a five-year progression free survival (PFS) that ranged from 25 to 72 months [118].

In a large, not-randomized study, 194 CUP patients received an assay-directed therapy according to the molecular prediction; in this case, the median survival of tumors predicted to be responsive to therapy was significantly longer than resistant (13.4 versus 7.6 months; *p* = 0.04) [112]. Similar evidence was collected from a prospective phase II clinical trial (NCT00936702), in which 46 CUP patients were enrolled and treated with carboplatin and paclitaxel. Thirty-eight of these patients were tested with a ResponseDX Tissue of Origin Test [102]: 19 of them were predicted as tumors sensitive to platinum/taxane therapy. Comparing the response to treatment of the two groups, significantly longer PFS (6.4 versus 3.5 months; *p* = 0.026) and overall survival (OS, 17.8 versus 8.3 months; *p* = 0.0052) in tumors predicted to be sensitive was observed [107].

In contrast to the above-mentioned studies, a site-specific therapy guided by molecular profiling failed to demonstrate its efficacy in a randomized prospective phase II trial, showing a one-year survival rate lower compared with conventional chemotherapy (44% versus 54.9%; *p* = 0.264) and no significant improvement in OS and PFS rates [129]. However, primary site prediction was based on a poorly validated molecular test [130], thus underlying the necessity of selecting an assay with reliable performance.

No encouraging results were reported by the European phase III clinical trial NCT01540058 (GEFCAPI 04), which included 243 CUP patients who underwent treatment with empiric chemotherapy combination (cisplatin and gemcitabine) or molecular gene expression profile-based therapies (Pathwork Tissue of Origin Test or CancerTYPE ID) [131]. However, results from the ongoing phase III clinical trial NCT03278600 could help clarify the real impact of tissue-of-origin profiling assays in predicting primary site and directing therapy in CUP patients.

Despite the high accuracy rates achieved by GEP assays, these tools are hindered by the low quality of RNA usually obtained from FFPE samples [132,133]. For decades, formalin fixation followed by embedding in paraffin has been the method of choice to preserve tissue samples; for this reason, this type of sample is the most commonly available source of specimens. However, both the fixation and embedding processes significantly impact RNA integrity and induce nucleotide chemical modifications [134,135]. Moreover, many other factors can impair RNA quality in FFPE samples: the time elapsed before the fixation; the thickness of the tissue; the temperature during fixation; and the storage time of FFPE samples [135]. For these reasons, in most GEP-based studies (summarized in Table 2), technical analysis failure caused by the low-quality of RNA obtained occurred in a variable portion of the enrolled patients.

### 4.2. Predictive Assays Based on Non-Coding RNAs

MicroRNAs (miRNAs) can be robustly detected irrespective of the quality of the sample due to their higher resistance to chemical modifications or degradation over time [136,137]. Therefore, miRNA expression was evaluated as a suitable alternative to gene expression to develop CUP site-of-origin predictive assays. Given the well-documented dysregulation of miRNAs in cancer, it is recognized that the expression of specific miRNA signatures can efficiently discriminate tumors from normal tissues and different cancer types [138,139,140,141,142].

Microarray-based studies explored the feasibility of a miRNA-based detection of the tissue-of-origin of tumor metastases, showing high accuracy rates (78–90%) when dealing with metastases of known origin [123,143,144]. Rosenfeld et al. [144] analyzed the profiles of 253 samples (up to 25 different histological subtypes) and built two classifiers by using decision-tree and K-nearest neighbors (KNN) algorithms. From this study, a 48-miRNA signature proved high accuracy to reveal the correct tissue-of-origin of metastases of known origin. This molecular assay was translated from microarray to RT-qPCR technology, preserving its analytical validity [144,145], to finally become the miRview mets assay (Rosetta Genomics, Princeton, NJ, USA) which is an LDT in USA [144].

When applied to a prospective series of 74 CUPs, molecular prediction was found to be consistent with clinicopathological features in 84% of cases [125]. Moreover, in a retrospective study, this assay was used to predict the tissue-of-origin of 57 patients, originally diagnosed as CUPs, characterized by central nervous system (CNS) metastases; predictions matched in 88% of cases with clinicopathological evaluation at the diagnosis, information obtained during the course of the disease, or specific examinations performed after and based on the results of the molecular assay [124].

To expand the number of tumor classes and include a wider range of carcinomas and neuroendocrine tumors, a second-generation assay named miRviews met^2^ or Rosetta Cancer Origin Test (Rosetta Genomics, Princeton, NJ, USA) was designed, reaching up to 42 different tumor types. Custom microarray data on a training set of 1282 samples were used to build a classifier with a combination of two different algorithms (decision-tree and KNN) that allowed tumor classification based on a signature of 64 microRNAs. The accuracy of this tool was 85% when applied on an independent blinded set of 509 primary tumors and metastases of known origin [146]. When applied on 84 CUP samples using this 64-miRNA signature, molecular prediction was found concordant with the first clinical hypothesis in 70% of patients, with treatment response in 89% of cases and with the final clinical diagnosis according to supplemental IHC stains in 92% of patients [127].

Interestingly, a microarray-based molecular study by Ferracin et al. [123] identified a 47-miRNA signature able to discriminate ten different cancer types with a prediction accuracy of 100% in primary cancers and 78% in metastases of known origin, which outperformed Rosenfeld’s signature when applied to the same dataset. Moreover, when applied to predict the primary site of 16 CUPs, predictions were found to be consistent with clinical and pathological hypotheses [123].

As confirmation of molecular testing performance, in a case regarding a 61-year-old female CUP patient with a suspected peritoneal or ovarian primary site, the molecular prediction by miRviews met^2^ (Rosetta Genomics, Princeton, NJ, USA) pointed out a mesothelioma origin; the prediction was later confirmed by additional IHC stains, leading to a therapeutic switch from platinum/taxane to pemetrexed/platinum salts that resulted in an improvement of the patient’s survival [127]. In another report, the clinical history of a 54-year-old male CUP patient was described; in this case the 64-mi, RNA assay predicted, with high probability (90%), a breast cancer origin, later confirmed by pathologic evaluation and IHC performed on breast resected metastasis, thus allowing optimal treatment for the patient [147].

### 4.3. Predictive Assays Based on DNA Methylation Profiling

Given its role as a regulatory mechanism of transcription, the use of DNA methylation patterns in CpG sites was also explored as a cancer biomarker. In a comprehensive study using the GoldenGate Assay (Illumina, San Diego, CA, USA), a DNA methylation fingerprint of 1628 human samples was obtained interrogating 1505 CpG sites. Fernandez et al. [126] analyzed 424 normal tissues, 150 non-cancerous disorders, and 1054 tumorigenic samples, including 855 primary malignancies (19 solid tumor types and 10 types of hematologic malignancies), 50 metastatic tumors, 25 premalignant lesions, 82 cancer cell lines, and 42 CUPs. Methylation patterns were found to be tissue-type specific, and it was demonstrated that, during tumorigenesis, human cancer cells undergo a progressive gain of promoter CpG-island hypermethylation and a loss of CpG methylation in non-CpG-island promoters. Comparing DNA methylation fingerprints of CUPs with primary and metastatic tumors with known origin, the investigators were able to assign a site-of-origin in 69% of CUP cases and create a prediction heatmap. Tumor type prediction was fully confirmed in 78% of cases in which pathological information was available [126].

Moran et al. [128] developed a classifier based on microarray DNA methylation signatures (EPICUP assay; FERRER, Barcelona, Spain) using a training set of 2790 known primary tumors, representing 38 tumor classes, and including 85 metastatic tumors. Analyzing 216 CUP samples, molecular prediction showed an overall accuracy of 90% when compared with the primary site unmasked during the disease follow-up or in post-mortem autopsy, or available pathological assessment. Moreover, analyzing available survival and therapy information of 114 CUP patients, a significantly improved OS of patients who received a tumor type-specific therapy (13.65 months) compared to those who received empiric treatments (sic months), could be observed [128].

In one in silico study, a new feature selection strategy was applied to build an miRNA- and methylation-based classifier, which proved to be more effective in the identification of predictive features (miRNAs or methylation level) than that in primary site prediction. For this purpose, 6602 samples with an available miRNA profile and 5379 samples with an available DNA methylation profile were collected from The Cancer Genome Atlas (TCGA), representing 14 tumor types. On independent datasets, the performance of these classifiers was found very promising, reaching an accuracy of around 95% for DNA methylation and 91% for miRNA [148]. However, this approach has not yet been tested on CUPs.

## 5. Driver Mutations and Precision Medicine

Our molecular understanding of cancer is rapidly expanding; therefore, histological cancer classification is gradually being replaced by a more informative molecular classification. The availability of public multi-omics database projects, such as The Cancer Genome Atlas (TCGA) [149], supports the shift towards a brand-new molecular classification of cancer, though shaping the future of cancer therapy [150,151]. With the revolutionary advent of personalized medicine, patient management is associated with the discovery of specific molecular characteristics on which therapeutic choices are made, in order to avoid the selection of suboptimal treatments.

In the IMPACT study (Initiative for Molecular Profiling and Advanced Cancer Therapy), patients with advanced cancers received matched targeted therapies (MTT) based on their mutational profile. Results from this study strongly support the employment of genomic matching: MTT-treated patients showed higher response rates (complete and partial response) (11% versus 5%; *p* = 0.0099), longer failure-free survival (3.4 versus 2.9 months; *p* = 0.0015), and longer overall survival (8.4 versus 7.3 months; *p* = 0.041). In addition, when treated with MTT, a significant difference was observed in OS between responders and non-responders (23.4 versus 8.5 months; *p* < 0.001) [152].

Assuming that the identification of CUP-specific druggable alterations could address the issue of their limited treatment options, several researchers have focused on the analysis of the CUP mutational landscape. High heterogeneity across different CUP patients was reported by several studies [153,154,155] (summarized in Table 3); of note, the most common alterations involve putative driver genes that are mutated in many different cancer types, mostly affecting core mitogenic, cell growth pathways (especially PI3K and MAPK pathways) and epigenetic deregulation [156,157,158].

In a minority of CUP patients, a common molecular signature was identified, mainly related to exogenous mutagens exposure, such as smoking or UV radiation [155,156]. In addition, mutational profiles seem to differ according to CUP histological subtypes: adenocarcinoma and poorly differentiated carcinoma subtypes exhibit mutations in genes involved in signal transduction pathways, while squamous cell carcinoma carry mutations in cell cycle control and DNA repair genes [154]. Chromosomal instability was suggested as a possible explanation for the unfavorable CUP subset [162]. However, other mechanisms that could explain CUP aggressiveness and early invasiveness are emerging.

A comprehensive retrospective analysis, using the 236-gene FoundationOne assay (Foundation Medicine, Cambridge, MA, USA), explored the genomic profiles of 200 CUPs [153]. At least one clinically relevant genetic alteration was found in 96% of CUPs, with a mean of 4.2 alterations per tumor. The most frequently mutated genes were *TP53* (55%), *KRAS* (20%), *CDKN2A* (19%), *MYC* (12%), *ARID1A* (11%), and *MCL1* (10%). According to this study, potentially druggable mutations were discovered in 20% of CUPs and, of note, alterations in the receptor tyrosine kinase (RTK)/Ras signaling pathway were more frequently detected in adenocarcinoma versus non-adenocarcinoma CUPs (72% versus 39%, *p* < 0.001) [153]. In addition, the amplification or mutation of *ERBB2* (10% versus 4%) and the alterations of *EGFR* (8% versus 3%) or *BRAF* (6% versus 4%) were predominantly observed in CUP characterized by adenocarcinoma histology [153]. *MET* was found to be mutated more frequently in CUP patients compared to metastases of known origin, suggesting its possible role to define CUP aggressive phenotype [163]. Of note, *MET* and *CTNNB1* mutations were found to be associated with a poor outcome (OS 11 versus 21 months, *p* = 0.015) [159].

At the Memorial Sloan Kettering Cancer Center, a comprehensive 468-gene panel called “MSK-IMPACT” was developed, which was used to analyze over 10,000 tumor samples, including 186 CUPs [164]. Mutational data from these patients, publicly available at cBioPortal [165], reported at least one mutation in 92% of CUPs, with the most frequently mutated genes in *TP53* (45.2%), *KRAS* (22.6%), *ARID1A* (14.5%), *SMARCA4* (12.9%), *KEAP1* (12.4%), *KMT2D* (9.7%), *CREBBP* (9.7%), *PIK3CA* (9.1%), *STK11* (8.6%) and *TERT* (8.6%); the most common CNAs were homozygous deletions of *CDKN2A* (10.2%) and *CDKN2B* (9.7%) and amplifications of *CCNE1* (4.3%), *ERBB2* (4.3%) and *MCL1* (4.3%) [164]. The MSK-IMPACT study reported alterations on the *TERT* promoter, which is associated with poor prognosis as a mechanism of apoptosis evasion [166,167].

These mutation rates are considerably comparable to those identified consulting the “GENIE” database, an AACR (American Association of Cancer Research) database project that collects whole exome sequencing data of almost 80,000 cancer patients, including 658 CUPs. CUP samples had at least one mutation in cancer-related genes, and the most common alterations were in *TP53* (40.6%), *KRAS* (17.9%), *ARID1A* (13.5%), *KEAP1* (10.7%), *KMT2D* (10.7%), and *SMARCA4* (10.6%); homozygous deletions of *CDKN2A* (11.9%) and *CDKN2B* (11.7%) and amplifications of *MYC* (4.8%) were the most frequent CNAs [168].

The variability in the identification of actionable alterations in CUPs across different studies (from 15 to 96%) (reported in Table 3) made it necessary to define more stringent criteria to assess actionability of genetic findings. Varghese et al. analyzed the frequency of druggable genetic alterations in 150 CUP samples tested with the MSK-IMPACT gene panel using OncoKB resources (a knowledge database that contains information about somatic alteration impact on function and their treatment implications (http://oncokb.org)) [169]. Among 150 CUP samples, 30% were found to harbor potentially druggable alterations (FDA level 2–3 of evidence for actionability), and 10% of them received matched targeted therapies [155]. Previous studies reported higher frequencies of druggable alterations, but this is explained by the inclusion of potentially actionable mutations that have no approved therapeutic agent (e.g., mutations in RAS family). Applying Varghese selection criteria to previously published datasets, frequencies of CUP cases potentially eligible for basket trials were found to be consistent.

The employment of biological rationale targeted therapy on CUP patients is still a matter of discussion, because the response rates of these drugs substantially vary across different cancer types. Thus, the selection of the right agent remains challenging. The knowledge of the primary site remains fundamental to improve CUP prognosis, because specific driver mutations could be predictive of responses in some tumor types but not in others. One example is represented by *BRAF* mutation (V600E), which is a predictive biomarker of response to BRAF inhibitors in melanoma but not in colorectal cancer [170].

In a prospective, randomized, phase II clinical trial (SHIVA study) a primary-independent application of targeted therapy on patients with metastatic solid tumors, including five CUPs, did not show any improvement on survival [171]. Results from the MOSCATO (Molecular Screening for Cancer Treatment Optimization) and ProfiLER trials showed that about 30% of genetically profiled tumors were suitable for targeted therapy, but only a small percentage (6–7%) of them were actually treated with this approach [172,173].

Anyway, genomic profiling of CUPs demonstrated a meaningful clinical relevance in several studies, and case reports guiding treatment decision-making might significantly affect the prognosis of these individual patients [95,174,175,176,177,178]. In a phase I trial, the mutational landscape of 17 CUP patients was explored, and seven of them were suitable for specific matched targeted therapies; of note, three patients showed stable disease, and one had a mixed response [158]. Despite these encouraging results, basket trials on metastatic solid tumors, including CUP patients, poorly succeeded to evaluate the impact of targeted therapy on CUPs.

However, the combination of molecular and genomic profiling to guide treatment decision was found to be a promising strategy, as reported by Hainsworth et al. [119]. According to this study, among 21 molecularly profiled CUP patients (miRNA 92-gene molecular cancer classifier assay, CancerTYPE ID) predicted as NSCLC, four harbored *EML4-ALK* fusion genes and one patient who was suitable for treatment with an ALK (Anaplastic Lymphoma Receptor Tyrosine Kinase) inhibitor showed a prolonged benefit. Similarly, results from the SUPER (Solving Unknown Primary Cancer) study on a prospective cohort of 172 CUP patients support the employment of a joined approach using mutational and GEP profiling to improve CUP management [179].

Proof-of-concept classifiers were developed by Marquard et al., using publicly available databases to infer tumor primary site from its genomic profile [180]. According to this study, an accurate prediction was obtained when employing a classifier based on point mutations and copy number alterations of 232 cancer-related genes and 96 classes of base substitutions, thus suggesting its potential utility for primary site detection in CUP patients. These classifiers are also freely available from the website interface TumorTracer [181].

The PCAWG (Pancancer Analysis of Whole Genomes) consortium developed a deep learning-based classifier of the primary site based on the analysis of somatic passenger mutations detected by whole genome sequencing of 2606 tumors, belonging to 24 cancer types [182]. Such an approach was able to reach an accuracy of 88% and 83% when applied on independent cohorts of primary and metastatic tumors, respectively. Such an approach has not yet been applied on CUP cohorts, although a hint of its potential could be assumed by a published case report on a 67-year-old male. This patient received a final diagnosis of malignant melanoma after three misdiagnoses, thanks to the combined application of next-generation sequencing (NGS) analysis, a patient-derived xenograft model, and tissue-of-origin prediction using TCGA transcriptomic data [183].

In their report, Penson et al. [184] applied a machine-learning approach for tumor type prediction using target DNA sequencing data. The algorithm was trained on a cohort of 7791 patients of advanced tumors belonging to 22 different classes, and later tested on an independent set of 11,644 cases. The prediction accuracy observed was 73.8% and 74.1%, respectively; moreover, when tested on plasma cell-free DNA, a similar accuracy (75%) was observed, suggesting a great potential of liquid biopsy for cancer diagnosis and primary site identification. In addition, Penson et al. [184] applied the algorithm for the prediction of the primary site on 141 CUP patients, obtaining a likely primary site in 67.4% cases. Such an approach, applied on two prospective patients with a first hypothesis of metastatic breast cancer, led to a final diagnosis and a change of treatment with an evident clinical benefit for these patients [184]. Results from ongoing clinical trials or basket trials extended to CUP patients could help clarifying the role of driver mutations in CUP patient management. In particular, in the ongoing clinical trial of Foundation Medicine (NCT02628379), CUP patients are treated according to their specific actionable mutations determined by FoundationOne CDx next generation sequencing (NGS) assay (Foundation Medicine, Cambridge, MA, USA), irrespective of their primary site.

Immunotherapy is emerging as a potentially winning therapeutic strategy in several cancer types; therefore, its possible administration on CUP patients has gained interest. Immunotherapy biomarkers of response are currently intensively studied, and to date they include programmed cell death ligand 1 (PD-L1) expression, high tumor mutational burden (TMB), and high microsatellite instability (MSI), which has been reported to occur in 28% of CUP cases [157,161,185]. NCCN guidelines support the use of pembrolizumab, an FDA-approved anti-PD1 antibody, on MSI-H or dMMR patients, regardless of the primary site [4]; in fact, this immune checkpoint inhibitor proved its antitumor activity in different cancer types, including colorectal cancer, melanoma, and non-small cell lung cancer [186,187,188,189]. However, CUPs with MSI-H or dMMR seem to be a minority [161]. Two published case reports describe CUP patients who received immunotherapy treatment with promising outcome, one with high PD-L1 expression and another with deficiency of the mismatch repair system or dMMR [174,190]. To date, several ongoing clinical trials aim to evaluate the impact of immunotherapy (pembrolizumab, nivolumab or Ipilimumab) on CUP survival (NCT03391973, NCT03752333, NCT04131621, NCT03396471, NCT02721732). Moreover, the impact on OS and PFS of novel targeted agents and immunotherapy on unfavorable CUP patients compared to chemotherapy may be highlighted from the results of the ongoing global phase II clinical trial of La Roche-Foundation Medicine (NCT03498521) [191].

### Liquid Biopsy

The urgent need to improve CUP diagnostic workup, paired with the critical health conditions of patients at the time of diagnosis, have led researchers to investigate the possibility to obtain information with less invasive procedures.

Liquid biopsy aims at analyzing tumor-derived components, e.g., circulating tumor cells (CTC) or circulating tumor DNA (ctDNA), in blood or other biological fluids, thus overcoming the many limits of conventional tissue biopsy. Indeed, the standard tissue biopsy is obtained by invasive procedures, usually only once at the time of diagnosis, and is sometimes not feasible due to poor tumor accessibility. Moreover, it is not representative of the entire intra-tumoral heterogeneity [192] and can be altered by a formalin fixation process that induces high levels of C > T/G > A transitions [193].

On the contrary, CTCs and ctDNA, flowing into the bloodstream from multiple metastatic sites, reflect inter- and intra-tumor heterogeneity of sub-clonal populations and provide a more reliable genetic and molecular information that could allow the application of personalized medicine. In addition, due to its reduced invasiveness, it enables the acquisition of serial samples during patients’ follow-up. For these reasons, liquid biopsy has the potential to be employed in CUP management, not only for primary site prediction, but also to estimate patient’s prognosis, response to treatment, minimal residual disease, and risk of relapse [15,194].

Several approaches were explored on common tumor types, based on the genetic and epigenetic profiling of ctDNA, staining of CTCs, and mRNA analysis of tumor-educated platelets (TEPs) for early detection studies or tissue-of-origin inference [195,196,197,198,199,200]. Another interesting approach was explored by Hoshino et al., who analyzed the diagnostic potential of proteomic profiles obtained from plasma-derived extracellular vesicles and particles (EVPs) [201]. Through the comprehensive analysis of EVP proteomes from 426 human cancer and non-cancer samples, they were able to select tumor-type specific EVP proteins as reliable biomarkers for early cancer detection and tissue-of-origin prediction. However, these proof-of-concept studies did not include any CUP patients, and so far, the application of liquid biopsy in the clinical management of CUP patients cannot be supported due to the low number of studies.

Liquid biopsy was firstly proposed for tissue-of-origin detection by targeting CTCs with a set of five fluorophore-conjugated antibodies (CK7, CK20, TTF1, ER, and PSA) able to distinguish the most common tumor types: lung, colon, breast, and prostate cancers. However, such an approach was found to be inadequate for poorly differentiated tumors [202]. Moreover, to demonstrate the potential utility of methylation profiles of cell-free DNA, a probabilistic method named CancerLocator was proposed as a non-invasive tool of primary site prediction, which showed promising results to determine not only the presence of a tumor but also the organ tissue [203].

CTC count obtained with an FDA-approved CellSearch System (Menarini Silicon Biosystems (Castel Maggiore (BO), Italy) is a prognostic biomarker in breast cancer and other tumor types. In metastatic patients, it is possible to find approximately 10 CTCs/mL of blood [204]. Few studies have evaluated CTC count in CUP samples using CellSearch system or immunofluorescent methods, reporting CTC presence in more than 50% of CUP patients [205,206]. Komine et al. [205] observed a median number of 31 CTCs in CUP patients that apparently decreases after chemotherapy treatment, thus suggesting its possible role as a predictive biomarker for treatment response [205].

Several studies focused on genomic analysis of ctDNA and included some CUP cases among other tumors types, showing high sensitivity rates in the identification of oncogenic and actionable alterations in CUPs [207,208,209]. The study conducted by Kato and colleagues was the first genomic analysis performed specifically on a large cohort of CUP patients (*N* = 442). From the analysis of 54–70 genes on ctDNA, they detected at least one alteration in 80% of CUPs, and the most mutated gene was *TP53* (37.8%); in addition, mutations were detected in genes involved in the MAPK pathway (31.2%), PI3K signaling (18.1%), and cell cycle (10.4%) [190]. This study proved the efficacy of liquid biopsy to monitor the dynamic changes of ctDNA during treatment in a case report of an 82-year-old male CUP patient. Based on the detection of *KRAS* G12D and *MLH1* R389W mutations in ctDNA, this patient was suitable for a combined treatment with nivolumab and trametinib, and reached a partial response after eight weeks from the first administration [190].

## 6. Conclusions

Despite the remarkable progress in the pathological assessment of metastatic tumors observed in the last decade, cancers of unknown primary remain an enigma which current diagnostic procedures cannot easily solve. There is still the need to identify more effective predictive biomarkers, which could help in identifying the CUP primary site and, consequently, facilitate therapeutic decisions. Overall, there is a rising consensus in recommending molecular profiling, either based on coding or non-coding transcripts or epigenetic modifications, for tissue-of-origin prediction. In our opinion, small non-coding RNAs and epigenetic modifications are particularly appealing. Such biomarkers could potentially endorse the access to more specific therapies and improve patients’ life expectancy. Liquid biopsy on CUP patients could help in unveiling druggable alterations using a noninvasive approach. Therefore, molecular diagnostics, combined with genetic profiling, might become the standard of care for future CUP management.

## Figures and Tables

**Table 1 cancers-13-00451-t001:** Initial diagnostic algorithm for cancer of unknown primary (CUP).

K7+/K20+	K7+/K20−	K7−/K20+	K7−/K20−
Transitional cell Pancreas	Lung (adenocarcinoma) Breast Ovarian Endometrial Gastric Cholangiocarcinoma Thyroid	Colorectal	Hepatocellular carcinoma Lung (neuroendocrine) Lung (squamous) Renal CellProstate

**Table 2 cancers-13-00451-t002:** Tissue-of-origin profiling assays applied on CUP patients.

Study Year and Reference	Study Type	Data Type	No. of CUP Patients Profiled/Enrolled	Type of Sample	Method Analysis	No. of Tumor Types	Tumor Type Included in the Reference Set	Most Predicted Tumor Types	Validation	CUP Prediction Accuracy	Impact on Clinical Outcomes with Prediction-Based Treatment
2005, [121]	Retrospective	mRNA	13	FF and FFPE	10,500-gene GEM	14	breast, CRC, gastric, lung, melanoma, mesothelioma, ovarian, pancreas, prostate, renal, cSCC, head and neck, testicular, uterine	ND	Clinicopathological features	84%	ND
2006, [113]	Retrospective	mRNA	48	FFPE	10-gene RT-qPCR;	6	lung, breast, colon, ovary, pancreas, and prostate	ND	unmasked latent primary tumor sites identified months to years later from the initial diagnosis	76%	ND
2008, [106]	Retrospective	mRNA	38/38	FFPE	1900-gene GEM; CupPrint^®^ (Agendia, Amsterdam, The Netherlands)	49	ND	lung (24%), CRC (18%), pancreas (16%), ovarian (11%)	Clinicopathological features and IHC	85%	ND
2008, [115]	Prospective/ retrospective	mRNA	104/120	FFPE	10-gene RT-qPCR	6	lung, breast, colon, ovary, pancreas, and prostate	CRC (49%), NSCLC (33%), pancreas (21%), ovarian (14%)	Clinicopathological features	61%	Improved OS of prospective cohort treated with site-specific regimen vs retrospective cohort treated with empiric chemotherapy
2008, [122]	Retrospective	mRNA	21	FFPE	1900-gene GEM; CupPrint^®^ (Agendia, Amsterdam, The Netherlands)	49	ND	colon (19%), small and large bowel (19%), ovarian (19%), breast (9%)	Clinicopathological features	67%	ND
2010, [105]	Retrospective	mRNA	16/21	fresh-frozen	1550-gene GEM; Pathwork^®^ TOO Test (Pathwork Diagnostic, Redwood City, CA, USA)	15	bladder, breast, colorectal, gastric, hepatocellular, kidney, non–small cell lung, ovarian, pancreatic, prostate, thyroid carcinomas, melanoma, testicular, germ cell tumor, non-Hodgkin’s lymphoma, and sarcoma	CRC (5), breast (4), ovary (3), lung (2), and pancreas (2)	Clinicopathological features and IHC	62.5%	ND
2011, [123]	Retrospective	miRNA	16	FFPE	microArray; 47-miRNA signature	10	breast, colon, endometrium, stomach/gastric, kidney, liver, lung, pancreas, prostate, and melanoma	gastric (31%), lung (25%), pancreas (19%), liver (12%), colon (6%), and kidney (6%)	Clinicopathological features	85%	ND
2011, [124]	Retrospective	miRNA	57/60	FFPE	48-miRNA RT-qPCR; miRNA	25	ND	ND	Clinicopathological features	ND	ND
2011, [125]	Prospective	miRNA	74/104	FFPE	48-miRNA RT-qPCR; miRNA	25	ND	CRC (17%), ovarian (8%); SCC (8%), pancreaticobiliary (13%), NSCLC (8%)	Clinicopathological features	84%	ND
2012, [126]	Retrospective	DNA methylation	42	fresh-frozen	DNA methylation array, 1505 CpG sites selected from 807 genes; GoldenGate assay (Illumina, San Diego, CA, USA)	29	bladder, breast, cervix, colon, endometrium, esophagus, ganglioneurom, glioma, head and neck, kidney, liver, melanoma, neuroblastoma, NSCLC, ovarian, pancreas, prostate, stomach, testis, ALL, AML, CLL, DLBCL, FL, MCL, mBL, MM, MDS/MPS, mixed lineage leukemia.	CRC (34%), NSCLC (17%), breast (17%)	Clinicopathological features	78%	ND
2012, [116]	Retrospective	mRNA	42	FFPE	92-gene RT-qPCR CancerTypeID^®^ (bioTheranostics, San Diego, CA, USA)	30	CRC, Lung-adeno/large cell, breast, HCC, ovary, pancreas, kidney, bladder, gallbladder, skin/squamous, melanoma, sarcoma, endometrium, testis, thyroid, stomach, mesothelioma, prostate, brain, lymphoma, uterine	selection of CUP predicted as CRC	Clinicopathological features	54–86%	Improved OS compared to historical controls; better outcome in more responsive predictive tumor types
2013, [114]	Prospective/retrospective	mRNA	144/149	FFPE	92-gene RT-qPCR; CancerTypeID^®^ (bioTheranostics, San Diego, CA, USA)	30	CRC, Lung-adeno/large cell, breast, HCC, ovary, pancreas, kidney, bladder, gallbladder, skin/squamous, melanoma, sarcoma, endometrium, testis, thyroid, stomach, mesothelioma, prostate, brain, lymphoma, uterine	CRC (15%), lung-adeno/large cell (10.5%), breast (8.8%), HCC (5.8%), ovary (5.2), pancreas (5.2%), kidney (4%), bladder (4%)	Clinicopathological features, IHC and unmasked latent primary tumor sites identified months to years later from the initial diagnosis	74–77%	ND
2013, [112]	Prospective	mRNA	252/289	FFPE	92-gene RT-qPCR CancerTypeID^®^ (bioTheranostics, San Diego, CA, USA)	30	CRC, lung-adeno/large cell, breast, HCC, ovary, pancreas, kidney, bladder, gallbladder, skin/squamous, melanoma, sarcoma, endometrium, testis, thyroid, stomach, mesothelioma, prostate, brain, lymphoma, uterine	biliary tract (21%); CRC (10%); NSCLC (7%); breast (5%); urothelial (11%); pancreatic (5%)	ND	ND	Improved OS compared to historical controls
2013, [127]	Retrospective	miRNA	84/93	FFPE	64-miRNA RT-qPCR; miRviews met2 (Rosetta Genomics, Princeton, NJ, USA)	42	ND	ND	Clinicopathological features and IHC	92%	ND
2015, [118]	Retrospective	mRNA	30/25	FFPE	92-gene RT-qPCR; CancerTypeID^®^ (bioTheranostics, San Diego, CA, USA)	30	CRC, lung-adeno/large cell, breast, HCC, ovary, pancreas, kidney, bladder, gallbladder, skin/squamous, melanoma, sarcoma, endometrium, testis, thyroid, stomach, mesothelioma, prostate, brain, lymphoma, uterine	carcinoma (40%) (of which 30% were germ cell tumors and 20% were neuroendocrine tumors), sarcoma (30%), melanoma (20%) and lymphoma (8%)	Clinicopathological features and additional IHC or genetic testing post molecular profiling	84%	Improved PFS compared to historical controls
2015, [108]	Retrospective	mRNA	49	FFPE	>25,000 genes, GEM	18	urinary (bladder), breast, cholangiocarcinoma, CRC, gastric, kidney, HCC, lung, neuroendocrine, ovary, pancreas, prostate, squamous cell, thyroid, melanoma, mesothelioma, sarcoma, testis	SCC (18%), lung (14%), kidney (10%), CRC (8%), breast (8%), cholangiocarcinoma (4%), mesothelioma (4%)	Clinicopathological features	78%	ND
2016, [128]	Retrospective	DNA methylation	216	fresh-frozen	DNA methylation array, 485,577 CpG sites; EPICUP assay (FERRER, Barcelona, Spain)	38	ND	NSCLC (21%), head and neck SCC. (10%), breast (9%), colon (9%), HCC (7%), pancreas (7%)	Latent primary, clinicopathological features, autopsy	87–100%	Improved OS compared to patients treated with empiric chemotherapy
2016, [107]	Prospective phase II	mRNA	38/46	FFPE	2000-gene GEM; Pathwork^®^ OO Test (Pathwork Diagnostic, Redwood City, CA, USA)	15	bladder, breast, colorectal, gastric, hepatocellular, kidney, non–small cell lung, ovarian, pancreatic, prostate, thyroid carcinomas, melanoma, testicular, germ cell tumor, non-Hodgkin’s lymphoma, and sarcoma	NSCLC (21%), CRC (18%), ovary (18%), pancreas (16%)	ND	ND	Improved OS in platinum-responsive tumor types
2016, [107]	Prospective phase II	mRNA	38/46	FFPE	2000-gene GEM; ResponseDX Tissue of Origin™ Test (Cancer Genetics, Rutherford, NJ, USA)	15	bladder, breast, colorectal, gastric, hepatocellular, kidney, non–small cell lung, ovarian, pancreatic, prostate, thyroid carcinomas, melanoma, testicular germ cell tumor, non-Hodgkin’s lymphoma, and sarcoma.	NSCLC (21%); CRC (18%); ovary (18%); pancreas (16%	ND	ND	Improved OS and PFS in platinum-responsive tumor types
2019, [117]	Retrospective	mRNA	12/15	FF and FFPE	Whole Transcriptome	40	ND	ND	Clinicopathological features	ND	ND
2019, [129]	Prospective phase II	mRNA	130	FF	22,000-gene GEM; mRNA	39	ND	pancreas (20.8%), gastric (20.8%), lymphoma (20%), urothelium (6.2%), cervix (5.4%), ovary (4.6%), biliary tract (3.8%)	ND	ND	No significant difference in PFS and OS of site-specific therapy compared to standard chemotherapy; prognostic value of the predicted sites-of-origin

Abbreviations: ND, not determined; GEM, gene expression microarray; FF, fresh frozen; FFPE, formalin-fixed paraffin-embedded tissue; OS, overall survival; PFS, progression-free survival; CRC, colorectal cancer; NSCLC, non-small cell lung cancer; HCC, hepatocellular carcinoma; ALL, acute lymphoblastic leukemia; AML, acute myeloid leukemia; CLL, chronic lymphocytic leukemia; DLBCL, diffuse large B-cell lymphoma; FL–follicular lymphoma; MCL, mantle cell lymphoma; mBL, monoclonal B-cell lymphocytosis; MM, multiple myeloma; MDS/MPS, myelodysplastic syndrome/myeloproliferative syndrome; cSCC, cutaneous squamous cell carcinoma.

**Table 3 cancers-13-00451-t003:** Studies analyzing the mutational landscape of CUP tumors.

Study	Analyzed Genes	Reference	Year	No. of CUP Patients	% of Samples with at Least One Alteration	% Potentially Targetable Alterations	Most Common Genetic Aberrations
Retrospective target sequencing and CNA analysis	701	[156]	2013	16	100	81	TP53 (62%), GNAS (25%), NOTCH1 (18%), NOTCH2 (18%), CDKN2A (18%), PIK3CA (18%), BRCA1 (18%), and STK11 (18%).
Retrospective metanalysis of mutational profiling, CNA analysis and protein expression profiling	47	[157]	2014	1806	ND	96	TP53 (38%), KRAS (18%), BRCA2 (11%), PIK3CA (9%), STK11 (6%); EGFR (17%) and ERBB2 (5%) for amplification.
Retrospective targeted sequencing of CTNNB1, MET, PIK3CA, KRAS and BRAF	5	[159]	2014	87	66	37	CTNNB1 (19.5 %), KRAS (10.2%), PIK3CA (7%), BRAF (4.5%), and MET (4.5%).
Retrospective target sequencing using FoundationOne assay (Foundation Medicine, Cambridge, MA, USA)	236	[153]	2015	200	96	20	TP53 (55%), KRAS (20%), CDKN2A (19%), MYC (12%), ARID1A (11%), MCL1 (10%), PIK3CA (9%), ERBB2 (8%), PTEN (7%), EGFR (6%), SMAD4 (7%), STK11 (7%), SMARCA4 (6%), RB1 (6%), RICTOR (6%), MLL2 (6%), BRAF (6%), and BRCA2 (6%).
Retrospective target sequencing using Oncomine Focus Assay and Cancer Hotspot v2 panel (Thermo Fisher Scientific, Waltham, MA, USA)	76	[154]	2018	21	81	52	TP53 (47%), KRAS (12%), MET (12%) and MYC (12%).
Retrospective target sequencing using MSK impact assay	468	[155]	2017	150	91	30	TP53 (70%), KRAS (35%), CDKN2A (30%), KEAP1 (23%), and SMARCA4 (22%)
Retrospective targeted sequencing and copy number analysis	50	[160]	2016	55	84	15	TP53 (55%), CDKN2A (22%), KRAS (18%), SMAD4 (11%), FGFR3 (9%), ATM (7%), and RB1 (7%).
Retrospective target sequencing using NGS CLIA-certified assay	-	[158]	2017	17	88	41	Impaired P signaling (47%), Epigenetic deregulation (47%), and impaired cell cycle control (47%)
Retrospective target sequencing using NGS CLIA-certified assay	592	[161]	2018	389	ND	22	TP53 (54%), KRAS (22%), ARID1A (13%), PIK3CA (9%), CDKN2A (8%), and SMARCA4 (7%)

Abbreviations: CNA, copy number alteration; OS, overall survival; PFS, progression-free survival; RFS, recurrence-free survival; N/E, not evaluated, no multivariate analysis conducted; NGS, next-generation sequencing.

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
