# Peer review of "Cancer of Unknown Primary: Challenges and Progress in Clinical Management"

_cancers, 2021, doi:10.3390/cancers13030451_

Round 1

Reviewer 1 Report

Molecular diagnostic based on coding and non-coding 2 RNAs and liquid biopsy in Cancer of Unknown Primary by Laprovitera and Ferracin et al.

My background relates to noncoding RNAs, the main focus of this manuscript, however, is cancer diagnosis. My comments are therefore mostly editorial.

There are two main points that I would suggest.

- The manuscript appears to be written by different authors, I guess at least four people have contributed to the text. The first part needs close scrutiny with respect to English style and grammar. The second part, focusing on miRNA and lncRNA, is well written and informative. The third part contains some confusing expressions (‘These mutations rates are considerably comparable to those identified….’), some appear to be translations from another language. The final part only needs little editorial input.

- The second point is minor and relates to the subtitles. ‘Introduction, Discussion and Conclusion’ are generic and certainly ‘Discussion’ should be replaced by more meaningful subtitles.

A few examples where editorial changes should be made. The list is not complete.

Line 21:               The lack of an identified tissue of origin…..

Line 22:               ‘life expectancy’

Line 192:             ‘embryogenesis’ should read ‘to the embryonic development of’

Line 223:            ‘mainly’ not ‘manly’

Lines 295-97:     sentence difficult to understand, consider rephrasing (‘month-to-year’ is not a commonly used expression)

Lines 320-23:     This sentence does not make much sense to me

Line 408:             ‘patterns’ should read ‘pattern’

Line 479:             Rephrase  along the line ‘A comprehensive 468 gene-panel …..’

Author Response

- The manuscript appears to be written by different authors, I guess at least four people have contributed to the text. The first part needs close scrutiny with respect to English style and grammar. The second part, focusing on miRNA and lncRNA, is well written and informative. The third part contains some confusing expressions (‘These mutations rates are considerably comparable to those identified….’), some appear to be translations from another language. The final part only needs little editorial input.

Reply: English language has been edited by a native English speaker. Please find all the corrections in the tracked version of the manuscript.

- The second point is minor and relates to the subtitles. ‘Introduction, Discussion and Conclusion’ are generic and certainly ‘Discussion’ should be replaced by more meaningful subtitles.

 Reply: We thank the reviewer for this observation. Discussion section has been removed.

Reviewer 2 Report

The authors have done a fairly good job of highlighting and describing the importance of cancers of unknown origin. The review is timely and will be informative to the scientific community. 

I have one suggestion that authors should increase the font size in the figs. to make it more legible, and also should proofread to check for structural errors in the sentences.

Author Response

I have one suggestion that authors should increase the font size in the figs. to make it more legible, and also should proofread to check for structural errors in the sentences.

Reply: We provided a more legible version of the graphical abstract. English language has been revised by a native speaker.